# KEAP1-NRF2 Interaction in Cancer: Competitive Interactors and Their Role in Carcinogenesis

**DOI:** 10.3390/cancers17030447

**Published:** 2025-01-28

**Authors:** Marina Oskomić, Antonija Tomić, Lea Barbarić, Antonia Matić, Domagoj Christian Kindl, Mihaela Matovina

**Affiliations:** Division of Organic Chemistry and Biochemistry, Ruđer Bošković Institute, 10000 Zagreb, Croatia; moskomic@irb.hr (M.O.); atomic@irb.hr (A.T.); lbarbar@irb.hr (L.B.); amatic@irb.hr (A.M.); domagojchristiankindl@gmail.com (D.C.K.)

**Keywords:** KEAP1, NRF2, oxidative stress, protein–protein interactions, cancer development

## Abstract

The Kelch-like ECH associated protein 1 (KEAP1)–nuclear factor erythroid-derived 2-like 2 (NRF2) pathway is one of the main regulators of the response to oxidative and electrophilic stress in cells. KEAP1 is the inhibitor of NRF2 activation and NRF2 is a transcription factor which is activated in the presence of oxidants and electrophiles that could damage the cells. Therefore, it has a protective role; however, it is also overactivated in cancer, where it expresses its dark side, protecting cancer cells from chemotherapeutics and radiation therapy. One of the means of NRF2 activation in cancer is through the binding of competitive protein interactors to KEAP1, which blocks the KEAP1-mediated ubiquitination of NRF2 and its degradation in the 26S proteasome. This review provides an overview of the KEAP1 competitive protein interactors identified thus far and explores their involvement in cancer development.

## 1. Introduction

An American Cancer Society report estimates the emergence of around 2 million new cancer cases in the US in 2024. The most prevalent forms are prostate cancer (29% cases) in men and breast cancer (32% cases) in women. They are followed by lung and colorectal cancer in both groups. These three types of cancer account for 48% of cancer cases in men and 51% in women. It is estimated that there were around 600,000 deaths from cancer in the US in 2024, mostly from lung, colorectal, and pancreatic cancer. Lung cancer mortality rates have dropped by 59% compared to the peak in 1990 for men and 36% compared to the peak in 2002 for women; however, it still causes more deaths than colorectal, breast, and prostate cancer together [1]. The development of targeted cancer therapies commenced in 1970s with the clinical trials of tamoxifen, an estrogen receptor antagonist, for breast cancer therapy and continued with the development of monoclonal antibodies, kinase inhibitors, photodynamic therapy, and antibody–drug conjugates (ADCs). The newest form of therapies include bispecific T-cell engagers (BiTES), oncolytic virus therapies, peptide receptor radionuclides. and chimeric antigen receptor T-cell (CAR-T) therapy [2]. The combination of targeted therapies with early detection has led to a decrease in the mortality of the majority of cancers; however, there is still a lot of room for improvement left. The KEAP1-NRF2 pathway is often dysregulated in cancer and represents a potential therapeutic target. NRF2 has a dual role in cancer: it prevents the initiation phase of cancer by protecting the cells from oxidative and electrophilic damage that can lead to genomic instability and DNA damage. However, in later stages of cancer, the constitutive activation of NRF2 promotes cancer progression by protecting cancer cells from the reactive oxygen species (ROS) induced cell death, enabling the detoxification of chemotherapeutics, promoting metabolic re-programming, and suppressing the immune response by reducing inflammation. Therefore, NRF2-targeted therapies in cancer should combine both NRF2 induction and inhibition, based on the NRF2 status in cancer [3].

## 2. KEAP1-NRF2 Signaling Pathway

The KEAP1-NRF2 (Kelch-like ECH-associated protein 1–nuclear factor erythroid 2-related factor 2) signaling pathway is the main regulator of the oxidative and electrophilic stress response in the cell [4]. NRF2 is a transcription factor that regulates the expression of more than 200 cytoprotective genes, encoding proteins involved in Phase I (drug oxidation, reduction, and hydrolysis), Phase II (drug conjugation), and Phase III (drug transport) detoxification, glutathione (GST)- and thioredoxin (TXN)-based antioxidant systems, carbohydrate metabolism and NADPH regeneration, lipid metabolism (fatty acid oxidation and lipases), heme and iron metabolism, transcription, and proteasome and autophagy regulation [5,6]. The protein levels of NRF2 in the cells in basal conditions are kept low through the action of its interactor KEAP1, which binds it in the Cul3-RBX1 E3 ligase complex, in which NRF2 is ubiquitinated and subsequently degraded in the 26S proteasome [7,8,9]. The ubiquitination of NRF2 is blocked in conditions of oxidative or electrophilic stress through the modification of reactive cysteines in KEAP1, which changes the conformation of the KEAP1-Cul3-RBX1 complex, putting NRF2 in an unfavorable position for ubiquitination. This results in the translocation of newly synthesized NRF2 to the nucleus and the activation of the transcription of cytoprotective genes regulated by NRF2 [10,11,12] (Figure 1). While NRF2 has a protective role in normal cells, including the prevention of the initiation of carcinogenesis, once cancer is already initiated, NRF2 reveals its dark side by protecting cancer cells from chemo- and radiotherapy, resulting in an increased activity of NRF2, which is linked to poorer prognosis in several types of cancer. Consequently, the KEAP1-NRF2 signaling pathway is the subject of intensive research, as it represents an attractive target for the treatment of oxidative stress-related diseases and conditions [13,14,15]. Specific somatic mutations in KEAP1 or NRF2 that lead to the constitutive activation of the NRF2 pathway have been identified in a variety of cancer types, including lung, head and neck, and esophageal cancers [16,17]. According to the cBioPortal cancer genomic database, in a curated set of non-redundant studies, KEAP1 gene mutations are found in around 3% of samples; the highest percentage of KEAP1 mutations is found in lung cancer (13–16%), lung adenocarcinoma (15%), and non-small-cell lung cancer (NSCL, 13%). Mutations in the NFE2L2 (NRF2) gene were found in around 2% of studies and the highest percentage of mutations was found in endometrial carcinoma (12%) and lung adenocarcinoma (11%). Mutations in the KEAP1 gene are spread through the entire coding region, while NRF2 mutations are most frequently found in the Neh2 region, which contains ETGE and DLGex sites for binding to KEAP1 (https://www.cbioportal.org/; accessed on 20 January 2025) [18]. These mutations confer a growth advantage to cancer cells and lead to chemoresistance [19]. Moreover, NRF2 activation promotes metabolic reprogramming in cancer cells, thereby enhancing survival and proliferation [20]. The dual role of NRF2 in cancer, acting as a tumor suppressor in normal cells and an oncogene in cancer cells, emphasizes the complexity of targeting this pathway for cancer therapy [21]. KEAP1 mutations have been associated with poor prognosis in non-small-cell lung cancer, further underscoring the need to better understand KEAP1 functions beyond NRF2 regulation [22]. Apart from mutations in KEAP1, NRF2, and CUL3, NRF2 overactivation in cancer is also caused by epigenetic silencing of KEAP1, NFE2L2 gene amplification, alternative splicing of NFE2L2 mRNA, increase in NRF2 expression by oncoproteins, KEAP1 cysteine modifications by oncometabolites, and binding of competitive protein interactors to KEAP1 or NRF2 [23]. As mentioned previously, the therapeutic targeting of NRF2 in cancer might involve both the inhibition and induction of NRF2, based on its activity in the cancer tissues. Strategies for the assessment of NRF2 status include the identification of single nucleotide polymorphisms (SNPs) in the NRF2 gene, like rs6721961, which correlates with NRF2 activity, and measuring the expression of NRF2-controlled genes in cancer biopsies [3]. Challenges in the therapeutic targeting of NRF2 in cancer are numerous because of the crosstalk between NRF2 and many other cellular pathways, some of which are going to be described further in the subsequent sections. While there are a lot of NRF2 inducers, some of which are already approved for the treatment of multiple sclerosis (dimethyl fumarate) and Friedrich ataxia (omaveloxolone), there are no specific inhibitors of NRF2 thus far. However, there are several strategies for the development of compounds that could inhibit NRF2 in cancer, including inhibitors of the NRF2-sMAF interaction and compounds with indirect impact on NRF2, including PI3K inhibitors and inhibitors of glutaminase (GLS) and glucose-6-phosphate dehydrogenase (G6PD) [14]. An additional challenge in the development of targeted therapies for the modulation of the KEAP1-NRF2 pathway is the appearance of resistance to therapy. Strategies dealing with this challenge are being developed, including the use of CRISPR-Cas9 screen targeting the druggable genome to discover proteins that are indispensable for the growth of cancer cells. One such study discovered the dependency of highly aggressive human lung adenocarcinomas, harboring KEAP1 mutations that overactivate NRF2, on solute carrier family 33 member 1 (SLC33A1), an endomembrane-associated protein involved in autophagy regulation, which can be used to develop targeted therapies for this type of cancer [24].

The BioGRID database includes data on 368 unique interactors of human KEAP1 (https://thebiogrid.org/115156/summary/homo-sapiens/keap1.html; accessed on 17 December 2024 [25]. We were interested in the interactors that bind the Kelch domain of KEAP1 and compete with NRF2 for binding. Proteins included in the database based only on the identification of the interaction by high-throughput interactome analysis were not considered. Selection was made giving the priority to proteins that have the ETGE or an ETGE-like motif in the amino acid sequence and whose interaction was confirmed and investigated by more than one independent group; however, we also took into account the list of already published, confirmed interactors of KEAP1 from the literature [26].

## 3. Sequestosome-1 (SQSTM1/p62)

p62, also known as Sequestosome 1 (SQSTM1), is a multifunctional scaffold protein that plays a crucial role in various cellular processes, including autophagy, the oxidative stress response, and inflammation [27,28]. The main function of SQSTM1 is the sequestration of damaged and misfolded proteins and organelles into aggregates prior to their degradation, and it is associated with various signaling pathways [29,30]. Therefore, it is considered a protein involved in the crosstalk between endocytosis and proteasomal and autophagosomal degradation [31,32,33]. It contains several protein interaction domains, including the C-terminal UBA domain that binds polyubiquitinated proteins to target them for proteosomal and lysosomal degradation [34].

SQSTM1 contains an STGE-binding motif similar to the NRF2 ETGE in a KEAP1-interacting region (KIR) important for its interaction with KEAP1. Thus, it directly interacts with the Kelch domain of KEAP1 via this motif. This mechanism allows SQSTM1 to outcompete NRF2 for binding to KEAP1 and, consequently, inhibits NRF2 ubiquitination and degradation, as shown by [35,36]. The binding affinity of SQSTM1 to KEAP1 can be significantly enhanced by phosphorylation [37]. Under stress conditions, such as oxidative stress, the phosphorylation of SQSTM1 promotes its recruitment to ubiquitinated substrates, facilitating selective autophagy and further enhancing NRF2 activation [38,39]. By preventing the degradation of NRF2, SQSTM1 enhances NRF2 activity, leading to an increased expression of antioxidant genes that protect against cellular damage [40,41]. However, this protective mechanism can have dual consequences in cancer biology. While low levels of NRF2 can prevent tumor initiation by promoting cell survival under stress, high levels may enable cancer cells to thrive and proliferate by conferring resistance to chemo-therapeutic agents [42]. Positive feedback between SQSTM1 and NRF2 therefore has an important function in cancer development and progression. In recent years, there has been an accumulation of evidence demonstrating the importance of SQSTM1 (p62) in a range of malignancies, as shown in Figure 2. The abnormal expression of SQSTM1 is closely associated with tumorigenesis and an unfavorable clinical course in various tumor types; this is thoroughly explained in [43]. For instance, it has been shown that high expression of SQSTM1 is associated with higher stages of epithelial ovarian cancer (EOC), especially in serous carcinoma, whereby a distinct expression subtype (CytoHigh/NucLow) is strongly associated with poorer overall survival and the presence of residual tumors [44].

In ovarian cancer, SQSTM1 is involved in mechanisms that confer resistance to chemotherapy. It has been shown that the accumulation of SQSTM1 activates the KEAP1-NRF2-ARE signaling pathway, increasing antioxidant gene expression and reducing oxidative stress-induced apoptosis [45]. This constitutes one mechanism by which ovarian cancer cells develop resistance to cisplatin. Furthermore, high levels of SQSTM1 encourage high activity of the NF-κB pathway, mediating prosurvival signals in a way that further promotes tumor development [46]. Autophagy deficiency in apoptosis-defective tumor cells leads to the accumulation of SQSTM1, which subsequently provokes a positive feedback cycle of ROS generation and further genomic instability and tumorigenesis. Ras-driven transformation enhances SQSTM1 accumulation, which in turn increases NF-κB activity, dampening ROS production and inhibiting tumor cell death [47]. In lung cancer, high levels of SQSTM1 were found in approximately 60% of human lung adenocarcinomas and in 90% of lung squamous-cell carcinomas [27]. For example, it has been found that a high expression of SQSTM1 is associated with a rise in the levels of c-Myc and activation of the pathways of mTORC1 signaling, events implicated in the growth of tumors and their progression [44,45]. The association of a high expression of SQSTM1 with poor prognosis is further supported by findings that have demonstrated its involvement in oncogenic signaling pathways through the activation of mTORC1, which may shift cellular metabolism toward anabolism—an advantageous state for tumor cells [46].

Moreover, the role of SQSTM1 in autophagy connects it to metabolic reprogramming in cancer cells, implying that it affects not just oxidative stress responses but also influences larger metabolic pathways that are crucial for tumor survival and growth [47,48]. Targeting SQSTM1 (p62) in combination with NRF2 inhibitors has emerged as a promising therapeutic strategy in the treatment of cancers, especially in those malignancies with high expression levels of both proteins [47]. The possibility to effectively disrupt the p62-NRF2 feedback loop that might stop pro-oncogenic signaling since both autophagy and the NRF2-KEAP1 signaling pathway have been reported to show oncosuppressive and pro-tumoral roles [49,50]. By targeting both proteins, the pathway’s inhibition could be more effective than that achieved when either protein is targeted alone, hence improving the overall efficacy. For example, SQSTM1 inhibition may reduce its competition with NRF2 for binding to KEAP1 and thus potentiate the action of NRF2 inhibitors. Indeed, the authors of [47] have shown that both SQSTM1 and NRF2 represent potential druggable targets in cancers overexpressing these proteins and that their combined inhibition might restore p53 oncosuppressor function. Additionally, small-molecule inhibitors such as K67 have been identified to selectively disrupt the interaction between phosphorylated SQSTM1 and KEAP1 without affecting the NRF2-KEAP1 interaction and were able to restore the E3-ligase adaptor activity of KEAP1 [51]. This enhances the ubiquitination and degradation of NRF2 in hepatocellular carcinoma cells, leading to the inhibition of cell proliferation and a reduced tolerance to anticancer agents. This and other targeted approaches, together with direct NRF2 inhibitors such as natural and small-molecular NRF2 inhibitors [37,52], might constitute a promising strategy against the chemoresistance mechanisms of cancer and an improved efficacy of standard therapies.

## 4. Minichromosome Maintenance 3 Complex Component 3 (MCM3)

MCM3 is a subunit of the MCM2-7 helicase complex, which is an essential component in DNA replication. MCM3 contains within its helix-2-insert (H2I) beta hairpin a DxETGE motif that mimics the high-affinity binding motif of NRF2 to Keap1 [53]. Because of this structural similarity, MCM3 competes with NRF2 for KEAP1 binding, thus modulating the KEAP1-NRF2 antioxidant response pathway. The KEAP1-MCM3 interaction occurs in both nuclear and cytoplasmic compartments, with a slight preference for the cytoplasm [53]. Competition between MCM3 and NRF2 for KEAP1 binding represents a mechanism coordinating DNA replication with cellular redox homeostasis. Changes within the cellular levels of MCM3 modulate the sensitivity of the KEAP1-NRF2 antioxidant response pathway. Experiments have shown that when the levels of MCM3 are downregulated, the KEAP1-NRF2 pathway exhibits reduced sensitivity to xenobiotic stress [53]. Although the precise mechanisms have yet to be completely deciphered, several lines of evidence suggest that the KEAP1-MCM3 interaction directly contributes to the regulation of genomic DNA. For instance, it was suggested that the binding of KEAP1 to MCM3 interferes with the loading of the MCM2-7 complex onto DNA and is one way in which replication fork progression may be slowed down as a response to redox fluctuations [53]. This would, in turn, slow down the replication fork, allowing enough time for the potential lesions in the template DNA to be repaired, thus avoiding replicative stress and its associated genotoxic effects. A recent study also suggested that MCMBP may inhibit the degradation of newly synthesized MCM3 by preventing its interaction with KEAP1 or other E3 ligases [54]. This supports the complexity of the coordination of DNA replication and redox homeostasis, suggesting that the stability of MCM3 itself may be regulated in response to oxidative stress. In cancer development and progression, the KEAP1-MCM3 interaction could play a significant role. Recently, high expressions of MCM3 in various cancers, such as head and neck squamous-cell carcinoma, colorectal, bladder, renal cell carcinoma, medulloblastoma, and ovarian cancer, correlated with malignant properties and cell proliferation, were determined in the literature [55,56,57,58,59,60]. Overexpression of the MCM3 protein has been linked to poor prognosis in thyroid tumors, glioma, salivary gland tumors, melanoma, and cervical cancer [61,62,63,64,65]. Sun et al. (2024) [55] overexpressed MCM3 in tumor tissues of HNSCC and found its overexpression to be associated with better prognosis. In their study, significant overexpression of MCM3 was demonstrated at the mRNA and protein levels and was associated with longer overall survival in patients with HNSCC, suggesting that it might act as an independent prognostic factor. For colorectal cancer, Zhou et al. (2020) [56] demonstrated that elevated MCM3 expression correlated with advanced tumor stage and promoted G1/S cell cycle progression, proliferation, migration, and invasion. Chen et al. (2022) [57] revealed through bioinformatic analysis that MCM3 expression was significantly correlated with histologic grade and tumor histology in bladder cancer patients. In renal cell carcinoma, Gao et al. (2020) [58] demonstrated that PLK1 promotes proliferation and suppresses apoptosis by phosphorylating MCM3. This mechanism indicates the way in which MCM3 can take part in cancer development due to its interaction with other proteins. Cao et al. (2022) [59] performed a systematic analysis of MCM3 in medulloblastoma, indicating that it may play a role in this brain cancer. In ovarian cancer, Li et al. (2021) [60] found that MCM3 was overexpressed in cancer tissues, and its high expression was associated with poor prognosis. Experiments by Mulvaney et al. [66] indicate that KEAP1 has the ability to ubiquitinate MCM3. This modification might influence MCM3 function or its interactions within the MCM2-7 complex and, consequently, influence DNA replication in cancer cells.

KEAP1, MCM3, and NRF2 interactions seem to be part of some complex regulatory mechanisms, and those interactions could be exploited in cancer therapy and perhaps against altered MCM3 to inhibit NRF2 function or DNA replication process by this protein in cancers [60,67]. These findings point out that MCM3 is an important feature in a number of cancers because it can act as both a therapeutic target and a prognostic marker. Since MCM3 is associated with various cancer types, this suggests the vital role this protein plays in DNA replication and cell proliferation, which underlies the very development and progression of cancer.

## 5. Partner and Localizer of BRCA2 (PALB2, FANCN)

PALB2 (Partner and Localizer of BRCA2) was first identified as a protein that interacts with breast cancer type 2 susceptibility protein (BRCA2), a well-known tumor suppressor associated with higher breast and ovarian cancer risk. PALB2 is a large protein (1186 amino acids, 130 kDa) found only in vertebrates. It was co-immunoprecipitated with BRCA2 from the lysates of several cell lines and was indispensable for BRCA2 localization in nuclear foci and its recruitment to double-strand breaks in U2OS cells [68]. Subsequent investigations showed that PALB2 serves as a crucial bridge between breast cancer type 1 susceptibility protein (BRCA1) and BRCA2, facilitating the formation of a functional BRCA complex necessary for the efficient repair of DNA damage, particularly through homologous recombination (HR). The BRCA1-PALB2 interaction is essential for the recruitment of DNA repair protein RAD51 homolog 1 (Rad51) to double-strand breaks and their HR repair [69,70].

PALB2’s self-association plays a crucial role in controlling its activity during HR. Monomeric PALB2 is more proficient in DNA binding and facilitating RAD51 filament formation, suggesting that the regulation of PALB2 self-interaction could be a potential target for modulating DNA repair mechanisms [71]. Heterozygous mutations in PALB2 have been linked to a higher risk of breast [72,73] and pancreatic cancer [74], while homozygous mutations cause Fanconi anemia, a rare genetic disorder characterized by genomic instability and increased cancer risk [75,76].

PALB2 directly interacts with the Kelch domain of KEAP1 through its ETGE motif, competing with NRF2 for binding. PALB2 overexpression promotes the nuclear accumulation and activity of NRF2 and decreases ROS levels in the cell, while siRNA-mediated depletion of PALB2 decreases the mRNA expression of several NRF2 target genes and increases ROS levels. Interestingly, PALB2 depletion did not significantly change total NRF2 protein levels, suggesting that PALB2, being a nuclear protein, prevents the degradation of nuclear NRF2 and its export to the cytosol by binding KEAP1 in the nucleus [77].

While the interaction of PALB2 with KEAP1 did not lead to PALB2 degradation [77], it was shown that PALB2 is ubiquitinated by the KEAP1-CUL3-RBX1 E3 ligase at specific sites. This ubiquitination prevents PALB2 from interacting with BRCA1, thus inhibiting HR repair during the G1 phase, when cells do not have a sister chromatid available for double-strand break repair [78].

While there are no studies specifically focusing on the PALB2-KEAP1 interaction in the context of cancer development, it is plausible that this interaction could play a role in promoting cancer cell survival, chemoresistance, and tumorigenesis by increasing the activity of NRF2 in cells overexpressing PALB2. However, KEAP1-PALB2 interaction could also have an impact on HR repair. It could affect the localization, stability, or availability of PALB2 for DNA repair processes and create a feedback loop where DNA damage signals interact with the antioxidant response, enabling cancer cells to balance genomic integrity with oxidative stress.

## 6. Inhibitor of Nuclear Factor Kappa-B Kinase Subunit Beta (IκB Kinase β, IKKβ, IKBKB)

IKKβ is the main canonical regulator of the NF-κB signaling pathway, which regulates diverse cellular processes, including immunity, inflammation, cell survival, and proliferation. The NF-κB family of transcription factors includes five proteins in mammals: p50 (processed from p105), p52 (processed from p100), RelA (p65), RelB, and c-Rel. They form various dimeric combinations to regulate gene expression. In non-activated cells, NF-κB dimers are bound by the IκB family of proteins in the cytosol, preventing their binding to the DNA and maintaining their predominant cytosolic localization. In the presence of activating stimuli, comprising a wide range of signals including cytokines (TNF-α, IL-1β), microbial products (LPS), DNA damage (UV radiation, genotoxic agents), and Toll-like receptor (TLR) and TCR signaling, IκB is phosphorylated at specific serine residues by a multiprotein complex also referred to as the IKK signalosome. The IKK signalosome is composed of IκB kinase α (IKK α), IKKβ, and NEMO (NF-κB essential modulator), also known as IKKγ, a regulatory subunit that is essential for the activation of IKKα and IKKβ. NEMO functions as a scaffold, bringing upstream signals to the IKK complex. Once IκB is degraded, NF-κB dimers are released, so they translocate to the nucleus where they regulate the expression of genes involved in inflammation, immunity, cell survival, and proliferation [79,80]. In line with its involvement in diverse cellular processes, the NF-κB signaling pathway is dysregulated in a number of diseases, including chronic inflammation, autoimmune, neurodegenerative, and cardiovascular disease, diabetes, and cancer. NF-κB is overactivated in cancer, inducing the expression of proliferative and anti-apoptotic genes that enable tumor survival [80]. Considering the large number of genes controlled by the NF-κB and KEAP1-NRF2 signaling pathways, there is considerable crosstalk between them in the cell stress response; however, most of it is out of the scope of this review.

Our primary focus is the investigation of the interaction between IKKβ and KEAP1. Unlike in the case of most of the other competitive interactors of KEAP1, the main outcome of their interaction is not the upregulation of NRF2 but the downregulation of IKKβ and subsequently of the NF-κB signaling pathway. IKKβ binds the Kelch domain of KEAP1 through the ETGE motif. The direct interaction of IKKβ with KEAP1 leads to IKKβ ubiquitination in the KEAP1-CUL3-RBX1 complex and its degradation in the proteasome. The depletion of KEAP1 in several human breast cancer cell lines leads to the overexpression of several NF-κB controlled genes, including IL-8, involved in tumor angiogenesis. Analysis of clinical breast cancer samples revealed an inverse correlation between a high expression of CUL3/KEAP1 and IKKβ, while a combination of low KEAP1 and CUL3 expression and high IKKβ expression was a predictor of poor survival [81]. It is worth noting that IKKβ contains a DLG motif 126 amino acids downstream from ETGE, so the mechanism of its ubiquitination by KEAP1 could be very similar to that of NRF2 ubiquitination. IKKβ interaction and KEAP1-mediated degradation was confirmed by another group; however, their findings indicate that IKKβ is degraded through the autophagy–lysosome pathway. They also showed that KEAP1 overexpression inhibits the TNFβ-mediated activation of NF-κB and IKKβ phosphorylation [82]. A correlation between the genetic inactivation of KEAP1-CUL3-RBX1 E3 ligase and IKKβ upregulation and overexpression of the NF-κB target gene was found in non-small-cell lung cancer (NSCLC). Analysis of eight NSCLC cell lines in comparison to normal cells from the bronchial epithelium revealed a correlation between increased protein levels of IKKβ and genetic loss of KEAP1, CUL3, and RBX1 loci and gain of IKKβ. These findings were confirmed by siRNA knockdown of KEAP1, CUL3, and RBX1 in non-malignant bronchial epithelial cells [83]. It was also found that inhibiting heat shock protein 90 (HSP90) with geldanamycin (GA) destabilizes IKKβ, enhancing its association with KEAP1 and leading to its autophagic degradation. Notably, this process is not inhibited by tert-butylhydroquinone (tBHQ), an electrophile that typically suppresses KEAP1-mediated degradation of NRF2. Additionally, a leucine-to-alanine mutation at position 353 (L353A) in the ubiquitin-like domain (ULD) of IKKβ was shown to destabilize the protein, enhance KEAP1 binding, and promote autophagic degradation. These findings suggest that KEAP1 plays a role in degrading structurally unstable IKKβ, thereby negatively regulating NF-κB signaling under proteotoxic stress conditions [84].

The NF-κB and KEAP1-NRF2 pathways balance oxidative stress and inflammation in normal and disease states. While NRF2 generally counteracts the pro-inflammatory effects of NF-κB, dysregulation of this crosstalk contributes to the pathogenesis of chronic inflammatory diseases, cancer, and neurodegenerative disorders. Understanding their interplay offers opportunities for developing more effective therapeutic strategies.

## 7. Dipeptidyl Peptidase 3 (DPP3, DPP III)

DPP3 is a peptidase that cleaves dipeptides from the N-termini of 4–8 amino acid-long peptides with relatively broad substrate specificity [85,86]. That and its ubiquitous presence indicate that it is involved in the final stages of protein turnover in the cells; however, there are indications that it also has a role in the regulation of blood pressure, pain, and inflammation [87]. In 2013, Hast et al. found that DPP3 interacts with the Kelch domain of KEAP1 protein through its ETGE amino acid motif, located on the unstructured loop of its upper domain (Figure 3), competes with NRF2 for binding to KEAP1, inhibits NRF2 ubiquitination, and promotes NRF2-dependent transcription [88], while Matić et al. (2021) established that biding of DPP3 to the Kelch domain is a two-step process, the first step being the release of the ETGE loop from the DPP3 protein body [89].

More than 25 years ago, Šimaga et al. found that DPP3 is overexpressed in ovarian and endometrial cancer compared to normal tissue [90], and in a 2003 study, they found that the level and activity of DPP3 were correlated with the aggressiveness of primary ovarian carcinoma [91]. This suggested a potential role of DPP3 in cancer development. Since then, DPP3 overexpression has been implicated in the development of several other types of cancer, but the exact mechanisms of its involvement are still largely unknown; however, there is strong evidence that DPP3 overexpression is correlated with the overactivation of NRF2 in lung [88] and breast cancer [92]. Hast et al. (2013) found that DPP3 copy number gain and mRNA overexpression positively correlated with NRF2 activity in squamous-cell carcinoma of the lungs with high NRF2 activity but lacking NRF2 stabilizing mutations. It was also shown that tumor-derived mutations in KEAP1 are hypomorphic with respect to NRF2 inhibition, and that DPP3 overexpression in the presence of these mutants enhances NRF2 activation. These findings support the competition model of NRF2 activation in lung cancer [88], while integrated bioinformatics analysis identified that overexpression od DPP3 in lung squamous-cell carcinoma is strongly associated with poor survival [93]. Lu et al. (2017) found that DPP3 is overexpressed in estrogen receptor-positive breast cancer and that the overexpression correlates with increased NRF2 controlled gene activity [92]. Interestingly, analysis of 98 whole-exome samples from breast cancer patients by an integrated bioinformatics approach showed that high expression levels of DPP3 correlated with poor survival of breast cancer patients and were an independent survival determinant, making DPP3 a putative prognostic biomarker in breast cancer [94]. Fu et al. (2024) also found that DPP3 expression is higher in breast cancer tissues than that in adjacent tissues by analyzing both the Cancer Genome Atlas (TCGA) database and clinical samples. They found that patients with a high expression of DPP3 have poor survival outcomes [95]. The clinical importance of this peptidase in cancer has also been underscored by its inclusion as a cancer signature gene in a six-gene model used for the diagnosis and prognosis of breast and lung cancers [96]. There are also reports on the overexpression and oncogenic functions of DPP3 related to poor survival in numerous other malignancies, including colorectal cancer, where DPP3 was found to target cyclin-dependent kinase 1 (CDK1) [97], multiple myeloma [98], and esophageal carcinoma [99]. Arora et al. highlighted DPP3’s role in esophageal squamous-cell carcinoma (ESCC), showing that its overexpression leads to an increased proliferation, apoptosis, and migration of ESCC cells, while its knockdown reduces these effects and sensitizes cells to oxidative stress and chemotherapy by downregulating NRF2 pathway proteins [100]. This suggests a critical role of DPP3 in ESCC and the DPP3/NRF2 axis as a target for overcoming chemoresistance in ESCC.

In conclusion, current data suggest that DPP3 plays a role in the progression and development of various cancers, and that an overproduction of DPP3 might directly enhance the antioxidant response by allowing newly synthesized NRF2 to escape from the KEAP1 complex, promoting cancer cell growth through metabolic reprograming and increased protection against chemotherapy and radiotherapy.

## 8. Serine/Threonine-Protein Phosphatase PGAM5, Mitochondrial (PGAM5)

Serine/threonine-protein phosphatase (PGAM5) is a mitochondrial protein belonging to the phosphoglycerate mutase family, which plays an important role in the regulation of mitochondrial dynamics and cellular responses to stress. PGAM5 interacts with KEAP1, promoting the dissociation of the KEAP1-NRF2 complex under oxidative stress conditions [101,102]. This interaction allows newly translated NRF2 to translocate into the nucleus and activate the transcription of genes responsible for antioxidant responses. The PGAM5 gene encodes two isoforms of protein, PGAM5-L and PGAM5-S, through alternative splicing. Both isoforms contain an N-terminal NXESGE motif necessary for binding to KEAP1. PGAM5-L has been confirmed to bind to KEAP1 through this motif, while the interaction of PGAM5-S with KEAP1 is less well-characterized [102]. This interaction has been validated by two different groups in several ways, including affinity purification coupled with mass spectrometry and co-immunoprecipitation of overexpressed KEAP1 and endogenous PGAM5 [101,103]. PGAM5 is ubiquitinated by a KEAP1-dependent E3 ubiquitin ligase complex, which targets PGAM5 for proteasome-mediated degradation [102].

Research indicates that PGAM5 is often overexpressed in various cancers, including hepatocellular carcinoma (HCC) and colorectal cancer (CRC), with its high expression correlating with poor prognosis [104]. PGAM5 promotes chemoresistance and enhances cell survival through anti-apoptotic mechanisms, specifically by stabilizing Bcl-xL, an anti-apoptotic protein [105]. Further, PGAM5 exhibits a duality in cancer, as it functions to facilitate both the survival of the cell and tumor development by enhancing the activity of NRF2 and impeding apoptosis while participating in mitochondrial quality control processes such as mitophagy [106]. In recent years, PGAM5 was associated with a newly described pathway of cell death known as oxeiptosis, which is also linked to cancer therapy [107,108]. The complex interplay between PGAM5, KEAP1, and NRF2 has been implicated in the regulation of mitochondrial retrograde trafficking, further highlighting its importance in cellular homeostasis and stress responses [105].

Moreover, the redox-sensitive nature of KEAP1 influences its interaction with PGAM5, potentially affecting the balance between cell survival and death in response to oxidative stress and cancer therapies [109].

## 9. Prothymosin Alpha (PTMA)

Prothymosin alpha is a small acidic protein that plays crucial roles in various cellular processes, including cell proliferation, apoptosis, and immune modulation. It is particularly abundant in lymphoid tissues and has been implicated in cancer progression due to its ability to promote cell survival under stress conditions [110]. PTMA has a dual role: intracellularly, the protein regulates the cell cycle and apoptosis, while extracellularly, it may have immunomodulatory effects [111].

A key interaction of PTMA is with KEAP1, the negative regulator of NRF2. PTMA directly interacts with the C-terminal region of KEAP1, composed of six Kelch repeats, using its ENGE motif for this purpose [112,113]. This binding is significantly enhanced by divalent cations, which stabilize the PTMA-KEAP1 complex and facilitate its function [114]. Some studies suggested that in the nucleus, PTMA increases the levels of the KEAP1 transcript while contributing to NRF2 protein degradation. This dynamic interaction reflects the possibility that PTMA indirectly influences NRF2 activity by modulating cellular processes related to oxidative stress responses and oncogenic cell signaling pathways [110,115]. Although specific studies that have directly linked PTMA to the disruption of the KEAP1-NRF2 interaction are limited, there is evidence to suggest that PTMA’s role in cellular responses to stressors could enhance NRF2 signaling under specific conditions [112]. The implications of these interactions can contribute to increased malignancy in several cancers that have been associated with higher levels of PTMA. For example, it has been shown that PTMA can activate NRF2 to induce metabolic reprogramming, thereby enhancing tumor growth and survival [116,117].

Furthermore, PTMA’s ability to modulate KEAP1 and NRF2 pathway positions it as a potential target for therapeutic strategies aimed at manipulating these interactions to combat cancer progression. Activating NRF2, PTMA could affect metabolic pathways critically involved in cancer cell survival and proliferation [118,119].

## 10. Protein Niban 2 (FAM129B)

The FAM129B or NIBAN2 protein, also known as MINERVA, is a member of the FAM129 protein family comprising three proteins (FAM129A, B, and C) present only in vertebrates. It contains a pleckstrin homology (PH) domain close to the N-terminus, which is important for its localization to plasma membrane, helix bundle domain, and flexible C-terminal, proline-rich region that contains phosphorylation sites and KEAP1-binding motifs [120]. It was identified as a phosphorylation target of B-Raf signaling in the WM115 cell line, derived from a primary human melanoma tumor that had a dysregulated B-Raf/MKK/ERK pathway, crucial for the proliferation and survival of melanoma tumor cells. FAM129B phosphorylation at the proline-rich C-terminus caused loss of its cell–cell junction localization and increased melanoma cell invasion through the collagen matrix [121]. FAM129B stabilizes cellular contacts and suppresses apoptosis, while its depletion increases the rate of apoptosis in HeLa cells. Reduced apoptosis contributes to tumor growth by making cells more resistant to signals that would normally induce cell death [122]. FAM129B has been identified as a key factor that promotes cancer cell invasion in non-small-cell lung cancer (NSCLC) by facilitating the phosphorylation of focal adhesion kinase (FAK), which upregulates Matrix metalloproteinase-2 (MMP-2) and Cyclin D1. Increased FAM129B expression has been associated with shorter survival in lung cancer patients [123]. FAM129B is also phosphorylated via the epidermal growth factor receptor (EGFR), which in turn activates RAS signaling, essential for the growth, survival, and invasion of cancer cells. In cancer, EGFR signaling often leads to excessive cell proliferation and enhances metastatic potential. In this process, FAM129B may act as a regulator that enables cancer cell survival, thereby playing a critical role in tumor invasion and metastasis [124]. FAM129B knock-out in mice leads to delayed wound healing, indicating that it plays a key role in tissue regeneration and cell proliferation during recovery from injury [125]. Cheng et al. (2019) first showed that FAM129B competes with NRF2 for binding to KEAP1, and its overexpression decreases NRF2 ubiquitination and increases ARE-dependent transcription. FAM129B contains both the DLG and ETGE motifs, so the authors propose that it binds the KEAP1 dimer with both motifs; however, the ETGE and DLG motifs in FAM129B are separated by only 7 amino acids, unlike NRF2, where there are 47 amino acids between the two motifs, so it is highly unlikely that a monomer of FAM129B could bind the KEAP1 dimer with both sites. It was also found that FAM129B silencing increases the sensitivity of breast cancer cells towards chemotherapeutic oxaliplatin, while an increased expression of FAM129B in clinical breast cancer correlates with poor prognosis [126]. Schmidlin et al. (2021) showed that FAM129B binds KEAP1 through the ETGE motif and increases NRF2 protein expression in the A375 melanoma cell line. They also showed that FAM129B-KEAP1 interaction drives metastasis in these cells through hyperactive BRAF signaling. Namely, FAM129B is constitutively phosphorylated in A375 cells through the action of the BRAF V600E mutant, so its cytosolic localization is maintained, making it available for binding to KEAP1. When the phosphorylation of FAM129B is inhibited, FAM129B localizes on the plasma membrane, while the levels of NRF2 and the invasion and migration potential of the A375 cells decrease [127]. Elevated expression of FAM129B has also been found in cardiomyocytes, where it protects against hypoxia/reoxygenation injury by activating the NRF2/ARE pathway (antioxidant response element pathway). This pathway reduces oxidative stress, inflammation, and apoptosis, supporting cell survival under stress. The mechanism likely involves NRF2 stabilization and activation through FAM129B, enhancing the transcription of NRF2-regulated antioxidant and protective genes. Although this study focuses on cardiomyocytes rather than cancer cells, it highlights the role of FAM129B in promoting NRF2 activity during oxidative stress, which may improve cancer cell resilience and survival [128].

FAM129B role in regulating the NRF2/KEAP1 pathway has also been observed in diabetic nephropathy via adipose stem cell exosomes. The results show that FAM129B can modulate NRF2 activity through interaction with KEAP1, activating antioxidant mechanisms in kidney cells under stress. These findings are relevant to cancer research, as cancer cells often employ similar protective mechanisms. It suggests that FAM129B may act as a biomarker and therapeutic target for diabetic nephropathy treatment, with emphasis on its role in oxidative stress and inflammation [129].

In conclusion, FAM129B has a critical role in regulating the KEAP1/NRF2 pathway in various cell types. It competes with NRF2 for KEAP1 binding and promotes NRF2 activation, which contributes to cell survival, stress resistance, and invasive potential, positioning FAM129B as a potential therapeutic target to control the oxidative stress response and chemoresistance in cancer.

## 11. APC Membrane Recruitment Protein 1 (AMER1, WTX)

APC membrane recruitment protein 1 (AMER1) was first identified as being frequently mutated or deleted in Wilms tumor, a pediatric kidney cancer; hence, it was named Wilms Tumor gene on the X (WTX). AMER1/WTX is a vertebrate-specific gene encoding a 1135-amino-acid protein with two coiled-coil (CC) domains, one proline-rich (PR) domain, and a nuclear localization signal (NLS) at its N-terminus. This protein plays a critical role during development through the regulation of the Wnt signaling pathway [130]. The Wnt signaling pathway is involved in numerous cellular processes, including proliferation, differentiation, and migration. It has important roles in embryogenesis and organogenesis and is crucial for the self-renewal and maintenance of adult stem cells. The dysregulation of Wnt/β-catenin signaling is associated with a spectrum of diseases. Hyperactivation can lead to oncogenesis, contributing to cancers such as colorectal cancer, while hypoactivation is linked to degenerative diseases like osteoporosis [131].

Wnt signaling is downregulated under basal conditions through the action of the β-catenin destruction complex. AMER1/WTX has a dual role in Wnt signaling. It interacts with the β-catenin destruction complex, which includes proteins such as APC, AXIN1, AXIN2, protein phosphatase PP2A, GSK3α, GSK3β, and CK1α, and directly binds β-catenin and its E3-ubiquitin ligase adaptor β-TrCP, promoting β-catenin ubiquitination and degradation. Thus, it acts as a negative regulator of the Wnt signaling pathway. However, AMER1/WTX can also activate Wnt signaling by promoting LRP6 phosphorylation. AMER1/WTX translocates to the plasma membrane in a phosphatidylinositol 4,5-bisphosphate-dependent manner, where it facilitates the phosphorylation of LRP6 by bringing together necessary kinases and scaffold proteins [132]. Both activities require AMER1/WTX localization to the plasma membrane [133]. WTX can also translocate to the nucleus, where it localizes to specific subnuclear structures, known as paraspeckles, suggesting a role in transcriptional regulation. AMER1/WTX binds to WT1, a zinc-finger transcription factor and also a Wilms tumor suppressor, and enhances WT1-mediated transcription of target genes. This finding suggests that it plays a significant role in nuclear pathways involved in the transcriptional regulation of genes associated with cellular differentiation and tumor suppression [134].

AMER1/WTX is widely expressed in normal tissues, including the kidney, stomach, colorectum, esophagus, breast, and liver; however, its expression is markedly downregulated in corresponding malignant tissues. Similar trends of AMER1/WTX mRNA loss were validated by ISH and qRT-PCR, particularly in gastric cancer. The consistent downregulation of WTX across multiple cancer types implies that it may act as a general tumor suppressor gene and could serve as a biological marker for various cancers [135].

The first indication that AMER1/WTX is a binding partner of KEAP1 came from the investigation of the β-catenin protein interaction network [136]. This interaction was fully characterized several years later, when it was determined that AMER1/WTX inhibits NRF2 degradation by increasing its dissociation from KEAP1 and stabilizes NRF2. This stabilization allows NRF2 to translocate to the nucleus, where it triggers the transcription of ARE-driven genes [137]. Recently, it was established that homocysteine (Hcy)-induced senescence in a mouse neuroblastoma cell line is accompanied by increased levels of β-catenin and KEAP1. Hcy enhances the interaction between KEAP1 and AMER1/WTX while reducing the interaction between β-catenin and AMER1/WTX, thereby decreasing β-catenin degradation. It also decreases the methylation of the KEAP1 promoter CpG island, increasing its transcription. These findings suggest that Hcy-induced neuronal senescence involves the KEAP1-β-catenin pathway, presenting potential targets for therapeutic intervention in conditions like Alzheimer’s disease [138].

Considering that WTX is a tumor suppressor whose expression in most cancers is lower than in corresponding normal tissue, it is unlikely that its potential involvement in cancer development is through the stabilization of NRF2. However, it is plausible that the WTX-KEAP1 interaction may impact cancer progression in tissues where its levels are increased. Additionally, as it is a nuclear protein, it is possible that WTX plays a role in the stabilization of NRF2 within the nucleus and that nuclear AMER1/WTX levels do not correlate with its total levels.

## 12. Cyclin-Dependent Kinase 20 (CDK20)

Cyclin-dependent kinase 20 (CDK20) or cell cycle-related kinases with cell cycle p42 (CCRKp42) belongs to the family of cyclin-dependent kinases (CDKs), a group of serine/threonine kinases that regulate various cellular processes, including cell division, proliferation, differentiation, and stress response [139]. CDK20 was identified as a CDK-activating kinase (CAK) when its activity was first detected in HeLa cells. It was considered a second mammalian CAK, the first one having already been established as CDK7, which is both CDK and CAK. CDK20 phosphorylated CDK2 and CDK6 in vitro; however, it was not confirmed whether it has CAK activity in the cells [140]. Subsequent research gave conflicting results about CDK20 CAK activity. Liu et al. (2004) showed that CDK20 (p42) phosphorylates CDK2 in vitro and that its downregulation impaired CDK phosphorylation, activity and cell growth of HeLa cells [141]. On the other hand, Wohlbold et al. (2006) could not detect CDK20 CAK activity and found that its depletion impairs proliferation of HCT116 and U2OS cells, but it did not cause growth arrest [142].

The exact role of CDK20 in the cell is not completely elucidated; however, its overexpression was found in several types of malignancies. Overexpression of CDK20 was found in glioblastoma tumor tissue and glioma cell lines compared to the normal tissue, while siRNA silencing of CDK20 inhibited growth and reduced CDK2 phosphorylation in U-373 MG and U-87 MG glioblastoma cells [141]. Transcription of CDK20 (CCRK) was activated by ligand-bound androgen receptor (AR) in hepatocellular carcinoma cells (HCCs), and CDK20 in turn upregulated β-catenin signaling, which upregulated the expression of the β-catenin target gene, AR, creating a cycle that induced tumor growth. They also showed that CCRK mRNA is overexpressed in around 70 % of HCCs and that higher levels of expression correlate with poor survival of the patients [143]. Depletion of CDK20 affects the proliferation of glioblastoma cells, an aggressive type of brain tumor. Inhibition of CDK20 significantly reduces glioblastoma growth through ciliogenesis, indicating that CDK20 is a potential therapeutic target for glioblastoma treatment [144].

CDK20 also promotes resistance to radiotherapy and chemotherapy by activating the NRF2 pathway in lung cancer. The interaction of CDK20 with KEAP1, confirmed by co-immunoprecipitation of endogenous proteins in HEK293T cells, enables NRF2 activation, helping tumor cells survive therapy-induced stress. This makes CDK20 a critical factor in cancer cell resistance to standard therapies. Given this mechanism, inhibition of CDK20 could potentially reduce lung cancer cells’ resistance to chemotherapy and radiotherapy, thereby improving treatment efficacy [145].

In conclusion, CDK20 has a driver function in various cancer-associated pathways, and even though the mechanisms of CDK20 involvement in cancer progression are not completely elucidated, the downregulation of CDK20 in cancer has a proven therapeutic potential [146].

## 13. Nestin

Nestin is a class VI intermediate filament protein with a molecular weight of approximately 240 kDa and is a marker for stem cells. Highly expressed in a wide range of tumors, it plays significant roles in cell structure, organization, and signaling during both development and in adult tissues [147]. Nestin was shown to bind KEAP1, in competition with NRF2 binding, and inhibits the ubiquitination and degradation of NRF2 [148]. Nestin regulates the antioxidant system by stabilizing NRF2 protein levels and subsequently upregulating NRF2-ARE signaling. The ESGE motif of Nestin is responsible for its competitive binding to KEAP1, which subsequently protects NRF2 from degradation [148]. Nestin is also targeted by NRF2, therefore establishing a positive feedback loop, since NRF2 increases the transcriptional and expression levels of Nestin. The interaction with Nestin, therefore, stabilizes NRF2 and elevates the transcription of antioxidant genes, improving cellular redox homeostasis [149].

Nestin expression is associated with different aspects of carcinogenesis and tumor progression in several types of cancer and, especially, in non-small-cell lung cancer. Nestin expression in patients with NSCLC, according to [150], is significantly associated with lymph node metastasis and lymphangiogenesis and hence probably with cancer spread and progression. In the case of NSCLC, participation in redox balance and the response to oxidative stress for Nestin is practically only supported by the investigation of [148], where this protein was demonstrated to take part in the regulation of cellular redox homeostasis via the KEAP1-NRF2 feedback loop. In gastric cancer, Nestin promotes cell viability and prevents apoptosis by modulating the KEAP1-NRF2 axis, thereby facilitating tumor proliferation and metastasis [149]. Bidirectional regulation of Nestin and NRF2 proves the complexity of redox homeostasis in cancer cells and points toward potential use of this pathway for purposes in the treatment of cancers [151,152].

Since Nestin expression is closely related to tumor malignancy, further studies on its regulation and functions may suggest new therapeutic targets that inhibit tumor growth and improve treatment outcomes in NSCLC and other cancers [153].

## 14. Other Proteins

Beyond the interactors discussed in previous sections, we identified additional proteins containing ETGE or ETGE-like motifs that interact with KEAP1 and may play a role in carcinogenesis. All proteins covered in this review are listed in Table 1.

FAM117B was initially identified as a KEAP1 interactor in a large-scale proteomic study. Co-immunoprecipitation of endogenous FAM117B with overexpressed KEAP1 and Kelch domain of KEAP1 showed that it binds KEAP1 in an ETGE-dependent manner [88]. A recent study confirmed the endogenous KEAP1-FAM117B interaction in gastric cell lines HGC-25 and AGS and showed that FAM117B decreases NRF2 ubiquitination, leading to NRF2 activation. FAM117B promotes gastric cancer cell proliferation and reduces their sensitivity to chemotherapeutic agents in an NRF2-dependent manner. These findings suggest that FAM117B contributes to gastric cancer progression and chemoresistance via the KEAP1-NRF2 pathway, highlighting its potential as a therapeutic target [154].

Deubiquitinase OTUD1 was identified as a KEAP1 interactor in a proximity-labeling biotin ligase assay performed in multiple myeloma cells. This ETGE-dependent interaction was validated by co-immunoprecipitation of endogenous proteins, along with KEAP1-mediated ubiquitination of OTUD1. KEAP1 mediated ubiquitination of OTUD1 had no influence on its levels, so the biological role of this interaction remained unclear [155]. Oikawa et al. (2022) further confirmed KEAP1-OTUD1 interaction and showed that KEAP1 K63 ubiquitination levels were elevated in OTUD1-deficient mouse embryonic fibroblasts, while overexpression of wild-type OTUD1 (but not an ETGE-lacking mutant) reduced KEAP1 ubiquitination in HEK293T cells. Additionally, OTUD1 suppresses TNFα-induced NF-κB activation, suggesting that this interaction may represent a crosstalk between the NRF2-KEAP1 and NF-κB pathways, warranting further investigation [156].

A novel KEAP1 interactor, DPP9, cleaves dipeptides from polypeptides with N-terminal Pro or Ala residues at position 2. Unlike DPP3, DPP9 is a serine peptidase with low sequence similarity to DPP3. DPP9 has both ESGE and ETGE motifs; however, it binds KEAP1 through the ESGE motif. The interaction of endogenous proteins was confirmed in several clear-cell renal cell carcinoma and kidney renal papillary-cell carcinoma cell lines. DPP9-KEAP1 interaction inhibits NRF2 degradation and enhances the expression of antioxidant proteins such as GPX4 and SLC7A11. These proteins prevent lipid peroxidation and ferroptosis, promoting the survival of clear-cell renal cell carcinoma (ccRCC) cells under stress. DPP9 KO in renal cell carcinoma 768-O cells led to the accumulation of ROS and decreased NRF2 induction by H_2_O_2_, but the effect was reversed by the expression of DPP9-WT, and not ΔESGE mutant, in DPP9 KO cells. DPP9 overexpression in renal cancer cells induced the expression of SLC7A11, which led to the protection of cells from ferroptosis. DPP9 KO cells were more sensitive to the ferroptotic inducer sorafenib, but the overexpression of WT DPP9 in KO cells protected the cells from ferroptosis. These findings indicate that overexpression of DPP9 might protect renal cancer cells from ferroptosis through the binding of DPP9 to KEAP1 and the upregulation of NRF2-controlled genes and that the inhibition of KEAP1-DPP9 interaction might be a therapeutic strategy to overcome this effect in renal cancer [157]. Similarly, in liver cancer, DPP9-KEAP1 interaction, confirmed by co-immunoprecipitation of endogenous proteins, reduces NRF2 degradation, increasing NQO1 expression and decreasing chemotherapy efficacy. Elevated DPP9 expression in liver cancer cells is associated with resistance to chemotherapy and poor outcomes, suggesting that targeting DPP9 could improve treatment efficacy [158].

Gankyrin (PSMD10) interacts with KEAP1 through two distinct motifs (ELKE and ENKE). This interaction was confirmed by co-immunoprecipitation of endogenous proteins from HEK293T cell lysates. KEAP1–gankyrin interaction stabilizes NRF2 by reducing the ubiquitination and degradation of NRF2. NRF2, in turn, upregulates gankyrin expression, creating a positive feedback loop. Elevated gankyrin and NRF2 levels are linked to poor prognosis and aggressive hepatocellular carcinoma (HCC). Inhibiting gankyrin may disrupt this loop, impair NRF2-driven chemoresistance, and increase HCC sensitivity to chemotherapy [159].

Finally, RPB5-mediating protein (RMP) interacts with KEAP1 through two E**E-like motifs, inhibiting NRF2 degradation and enhancing NRF2 target gene expression. This activation supports cholangiocarcinoma cell proliferation, survival, and metastasis by creating a favorable redox environment. Elevated RMP expression is associated with increased tumor growth in vivo and poor prognosis in cholangiocarcinoma patients [160].

**Table 1 cancers-17-00447-t001:** Biochemically confirmed competitive protein interactors of KEAP1 with putative involvement in cancer progression through the activation of NRF2 activity.

Protein	Uniprot ID	Binding Motif	Cancer
Sequestosome-1 (SQSTM1, p62)	Q13501	STGE	Prostate [36,48], bladder [37], lung [38], liver [40,44,45], ovarian [42]
DNA replication licensing factor MCM3	P25205	ETGE	Not specified
Partner and localizer of BRCA2 (PALB2, FANCN)	Q86YC2	ETGE	Not specified
Inhibitor of nuclear factor kappa-B kinase subunit beta (IKKβ, IKBKB)	O14920	ETGE	Breast [81]; lung [83]
Dipeptidyl peptidase 3 (DPP3, DPP III)	Q9NY33	ETGE	Lung [88], breast [92]
Serine/threonine-protein phosphatase PGAM5, mitochondrial	Q96HS1	ESGE	Colorectal [104,108], ovarian [107], prostate [109]
Prothymosin alpha (PTMA)	P06454	ENGE	Bladder [118]
Protein Niban 2 (NIBAN2, FAM129B)	Q96TA1	ETGE	Breast [126]
APC membrane recruitment protein 1 (AMER1, WTX)	Q5JTC6	ETGE	Kidney [137]
CDK20	Q8IZL9	ETGE	Lung [145]
Nestin (NES)	Q6P5H2	ESGE	Lung [148], gastric [149]
Protein FAM117B	Q6P1L5	ETGE	Gastric [154]
OTU domain-containing protein 1 (OTUD1)	Q5VV17	ETGE	Not specified
Dipeptidyl peptidase 9 (DPP9, DPP IX)	Q86TI2	ESGE ^1^	Renal [157], liver [158]
26S proteasome non-ATPase regulatory subunit 10 (PSMD10, Gankyrin)	O75832	ELKE/ENKE ^2^	Hepatocellular [159]
Unconventional prefoldin RPB5 interactor 1 (URI1, RMP)	O94763	E**E ^3^	Cholangiocarcinoma [160]

^1^ DPP9 contains ETGE and ESGE motifs; however, it binds KEAP1 through ESGE motif located on C-terminal end. ^2^ Protein has ELKE motif on N-terminal and ENKE motif on C-terminal part. Deletion of either N- or C-terminus decreases binding. ^3^ Protein has two E**E motifs similar to ETGE, through which it binds KEAP1.

## 15. Conclusions

The KEAP1-NRF2 pathway is a critical regulator of cellular responses to oxidative and electrophilic stress. Its dysregulation, often through the disruption of KEAP1-NRF2 interaction, has been implicated in the later stages of carcinogenesis. Competitive interactors of KEAP1, including proteins that bind to KEAP1 or NRF2 and block their interaction, represent a significant mechanism for NRF2 activation. While the activation of NRF2 can provide cytoprotective effects that prevent cancer initiation, its persistent overactivation in established tumors contributes to cancer cell survival, metabolic reprogramming, and resistance to therapy.

Since NRF2 controls more than 200 genes and its downregulation can have deleterious effects on normal cells, targeting the specific interactions with proteins that compete with NRF2 for KEAP1 binding might prove to be a more effective strategy of NRF2 downregulation in some cancers. Understanding the molecular mechanisms and structural basis of KEAP1 competition by these interactors sheds light on their roles in promoting or inhibiting carcinogenesis and opens new avenues for therapeutic interventions. Specifically, targeting the KEAP1-NRF2 axis could have dual applications: reactivating KEAP1 to suppress NRF2 hyperactivation in advanced cancers or modulating NRF2 activity to protect normal tissues from oxidative damage during cancer therapy.

## Figures and Tables

**Figure 1 cancers-17-00447-f001:**
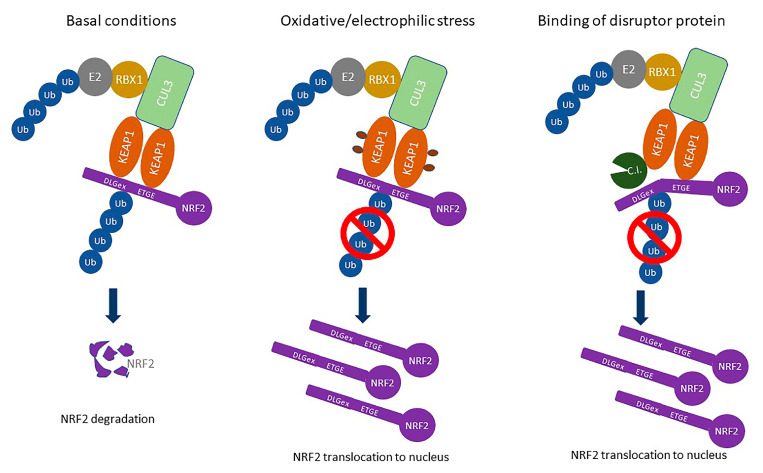
Regulation of the KEAP1-NRF2 signaling pathway. In basal conditions, NRF2 is bound in the KEAP1-CUL3-RBX1 E3 ligase complex through the interaction with the KEAP1 dimer, where it is ubiquitinated and subsequently degraded in the 26S proteasome. Under conditions of oxidative/electrophilic stress, reactive cysteines in KEAP1 are oxidized, the conformation of the E3 ligase complex changes, and NRF2 is no longer in the appropriate position for ubiquitination, which leads to the accumulation of the newly synthesized NRF2 and its translocation to the nucleus. The same effect is achieved by binding of the disruptor protein (C. I. for competitive interactor), which displaces the NRF2 DLGex site from the KEAP1 monomer and blocks NRF2 ubiquitination.

**Figure 2 cancers-17-00447-f002:**
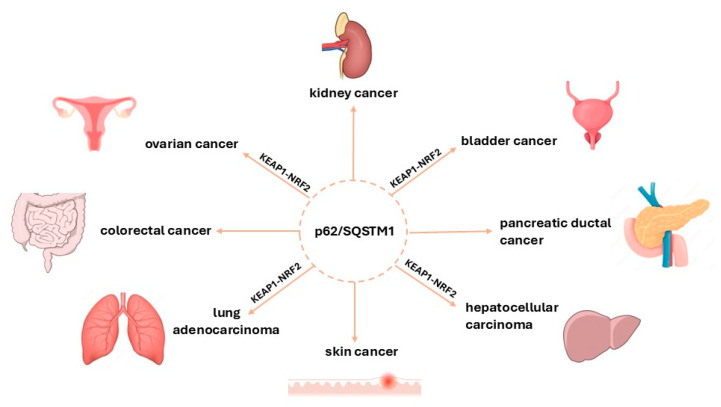
Involvement of SQSTM1 in different cancer types (carcinoma) and its interaction with the KEAP1-NRF2 pathway contributing to cancer development and progression. Schematic representation of the involvement of SQSTM1 in cancer, highlighting its importance as a target in potential cancer research and therapy. Notably, the interaction between SQSTM1 and KEAP1 is crucial in hepatocellular carcinoma, lung adenocarcinoma, and bladder and ovarian cancers. The interactions between SQSTM1, KEAP1, and NRF2 highlight the complex role of SQSTM1 in modulating oxidative stress responses via the KEAP1-NRF2 pathway for tumor survival and drug resistance.

**Figure 3 cancers-17-00447-f003:**
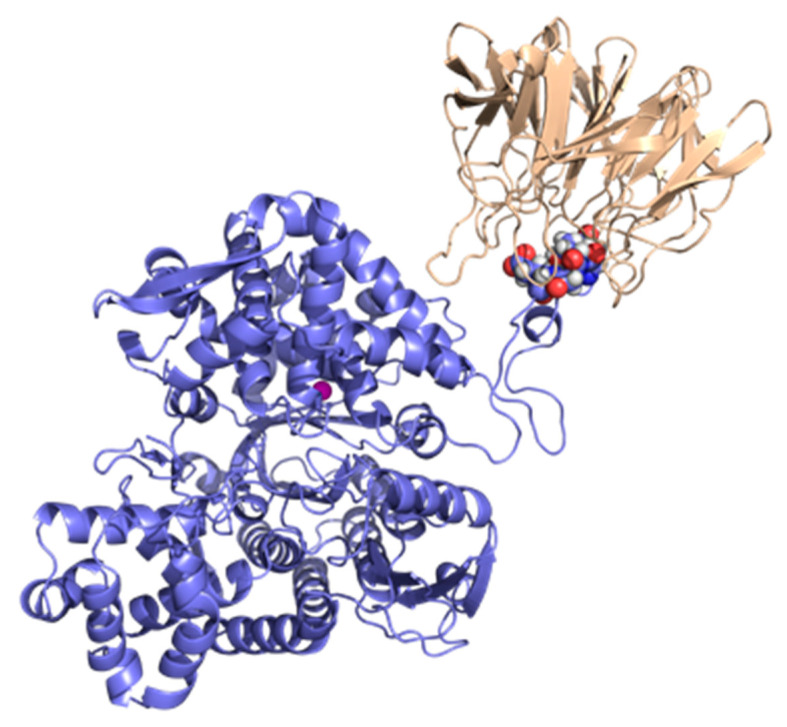
The DPP3–Kelch complex structure obtained by a comprehensive computational study (data not yet published; for details, see Appendix A). DPP3 is shown in violet, with ETGE motif residues are represented as spheres and carbon atoms, also colored violet. The Kelch domain is colored wheat, and the zinc ion is depicted as a magenta sphere.

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
