# Peer review of "KEAP1-NRF2 Interaction in Cancer: Competitive Interactors and Their Role in Carcinogenesis"

_cancers, 2025, doi:10.3390/cancers17030447_

Round 1
Reviewer 1 Report
Comments and Suggestions for Authors
The manuscript by Marina Oskomić and co-authors presents a well-structured review of the KEAP1-NRF2 pathway and its role in cancer development. The content is scientifically sound and provides valuable insights into this important cellular regulatory mechanism. The manuscript has notable strengths. The simple summary effectively introduces the topic to a broad audience, making complex concepts understandable. The organization flows logically, leading a reader through a comprehensive coverage of the KEAP1-NRF2 pathway and its dual role in cancer. The main analysis is focused on the role of competitive protein interactors that can bind to KEAP1, thereby disrupting its interaction with NRF2. The literature cited is up-to date and relevant. The manuscript deserves publication.
I have just the following minor comments.
Figure 1: Some symbols in the picture are too small and of low contrast.
Section 6 "Dipeptidyl peptidase 3 (DPP3, DPP III)": Considering that the structural peculiarities of the DPP3-Kelch complex are not yet published and are presented in this review for the first time, more discussion is needed concerning Figure 3.
In my opinion, the manuscript can be accepted for publication after minor revision.
Author Response
Comment 1: Figure 1: Some symbols in the picture are too small and of low contrast.
Response 1: We have changed the colours on the picture to enhance the contrast.
Comment 2: Section 6 "Dipeptidyl peptidase 3 (DPP3, DPP III)": Considering that the structural peculiarities of the DPP3-Kelch complex are not yet published and are presented in this review for the first time, more discussion is needed concerning Figure 3.
According to the reviewer’s suggestion, we revised the caption of Figure 3 from:
“Figure 3. The most representative structure of the DPP3-Kelch complex obtained through cluster analysis of microsecond-long MD simulations of various starting complex structures (data not yet published). DPP3 is shown in violet, with ETGE motif residues represented as spheres and carbon atoms also colored violet. The Kelch domain is colored wheat, and the zinc ion is depicted as a magenta sphere.”
to:
“Figure 3. The DPP3-Kelch complex structure obtained by a comprehensive computational study (data not yet published, for details, see Supplement). DPP3 is shown in violet, with ETGE motif residues represented as spheres and carbon atoms also colored violet. The Kelch domain is colored wheat, and the zinc ion is depicted as a magenta sphere.”
The following text was included in the Supporting Information:
The exact three-dimensional structure of the DPP III–Kelch complex has not yet been experimentally determined. However, several combined experimental and computational studies have been conducted to provide insights into the structure of the complex (in both of which members from our laboratory participated):
- Gundić, A. Tomić, R. C. Wade, M. Matovina, Z. Karačić, S. Kazazić, S. Tomić; Human DPP III–Keap1 Interactions: A Combined Experimental and Computational Study, Croatica Chemica Acta 89(2) (2016): 217–228.
- Matić, I. Kekez, M. Tomin, F. Bogár, F. Šupljika, S. Kazazić, M. Hanić, S. Jha, H. Brkić, B. Bourgeois, T. Madl, K. Gruber, P. Macheroux, D. Matković-Čalogović, M. Matovina, S. Tomić; Binding of Dipeptidyl Peptidase III to the Oxidative Stress Cell Sensor Kelch-like ECH-associated Protein 1 is a Two-step Process, Journal of Biomolecular Structure and Dynamics, 39(18) (2020): 6870–6881.
In the latter study, Small-Angle X-ray Scattering (SAXS) analysis and Hydrogen/Deuterium Exchange (HDX) mass spectrometry experiments provided insights into the overall shape of the complex and the regions of the proteins involved in the interactions, respectively. MD simulations of the complex in both studies were performed on the scale of several hundred nanoseconds.
With advancements in computational technology enabling MD simulations on increasingly longer timescales, as well as computational tools that facilitate protein-protein structure predictions (e.g., AlphaFold-Multimer and AlphaFold3), we extended our computational study on this subject, aiming to determine the most reliable complex structure. This was achieved by combining longer, microsecond-scale MD simulations with protein-protein structure prediction results. More specifically, several structures of the DPP III–Kelch complex were built based on our previous findings and AlphaFold predictions and subjected to numerous microsecond-long MD simulations using program Amber 22. Final trajectories were analyzed via clustering to identify the most populated complex structure, which is assumed to represent the most energetically favorable conformation. This structure is depicted in Figure 3.
Reviewer 2 Report
Comments and Suggestions for Authors
KEAP1-NRF2 signaling pathway is a critical cellular mechanism that protects cells from oxidative stress and electrophilic damage. While protecting normal cells from the initiation of carcinogenesis, NRF2 can also protect cancer cells from chemo- and radiotherapy. The authors focused on the competitive protein interactors of KEAP1 and reviewed their involvement in cancer development through blocking the KEAP1-mediated ubiquitination of NRF2. KEAP1 binds NRF2 in the CUL3-RBX1 E3 ubiquitin ligase complex, leading to its ubiquitination and degradation. By disrupting this interaction, some proteins can activate NRF2 in cancer and enhance its activity. Eleven proteins are discussed as the most important disruptors of KEAP1 in this manuscript, i.e., p62, MCM3, PALB2, IKKβ, DPP3, PGAM5, PTMA, FAM129B, AMER1, CDK20, Nestin. While the manuscript focuses on the roles of these interactors in cancer, it lacks an integrated overview of how these proteins collectively influence the KEAP1-NRF2 pathway.
1. I’m curious about the criteria or methods used to identify the 11 interactors as the most important ones in this manuscript.
2. Are all these direct interactions well established? It would be helpful to include the methods used to confirm these interactions in the manuscript.
3. What are the affinity values of these proteins for KEAP1?
4. Table 1 is not referenced in the main text and should be cited to ensure clarity and relevance.
Author Response
KEAP1-NRF2 signaling pathway is a critical cellular mechanism that protects cells from oxidative stress and electrophilic damage. While protecting normal cells from the initiation of carcinogenesis, NRF2 can also protect cancer cells from chemo- and radiotherapy. The authors focused on the competitive protein interactors of KEAP1 and reviewed their involvement in cancer development through blocking the KEAP1-mediated ubiquitination of NRF2. KEAP1 binds NRF2 in the CUL3-RBX1 E3 ubiquitin ligase complex, leading to its ubiquitination and degradation. By disrupting this interaction, some proteins can activate NRF2 in cancer and enhance its activity. Eleven proteins are discussed as the most important disruptors of KEAP1 in this manuscript, i.e., p62, MCM3, PALB2, IKKβ, DPP3, PGAM5, PTMA, FAM129B, AMER1, CDK20, Nestin. While the manuscript focuses on the roles of these interactors in cancer, it lacks an integrated overview of how these proteins collectively influence the KEAP1-NRF2 pathway.
Comment 1. I’m curious about the criteria or methods used to identify the 11 interactors as the most important ones in this manuscript.
Response 1:
Obviously, KEAP1 competitive interactors are not equally relevant to the pathophysiology of cancer. While some of these interactors were identified over a decade ago and have been studied by multiple groups (e.g., SQSTM1, MCM3, PALB2, IKKβ, DPP3, FAM129B), this does not necessarily imply greater relevance. Our primary criterion for inclusion was the identification of an interactor competing with NRF2 for binding to the Kelch domain of KEAP1, validated by more than one low-throughput method or independent research group. The main source of data for selecting these interactors was the BioGRID database, but we also considered already published, confirmed interactors of KEAP1 from the literature (ref. 25 in revised manuscript).
Following text was added to the manuscript:
The BioGRID database includes data on 368 unique interactors of human KEAP1 (https://thebiogrid.org/115156/summary/homo-sapiens/keap1.html; accessed in December 2024) [25]. We were interested in the interactors that bind Kelch domain of KEAP1 and compete with NRF2 for binding. Proteins included in the database based only on the iden-tification of the interaction by high throughput interactome analysis were not considered. Selection was made giving the priority to proteins that have ETGE or ETGE-like motif in the amino acid sequence, whose interaction was confirmed and investigated by more than one independent group, however, we also took into the account the list of already pub-lished, confirmed interactors of KEAP1 from the literature [26]
We acknowledge that this approach is not without limitations, and we may have underestimated certain newly identified interactors for which BioGRID data is incomplete (e.g., DPP9). Nevertheless, we believe this review provides a valuable and up-to-date overview of KEAP1 competitive interactors with potential roles in cancer development. We added a reference to the BioGRID database in the manuscript.
Comment 2. Are all these direct interactions well established? It would be helpful to include the methods used to confirm these interactions in the manuscript.
Response 2:
Most of the interactions described in the paper are confirmed and investigated by more than one group in different cell lines, therefore, we consider them well established and we think that it is not necessary to provide the details for the methods used to confirm the interaction. Those proteins are SQSTM1, MCM3, PALB2, IKKβ, DPP3, PTMA, FAM129B, Nestin and WTX. However, we agree that the disclosure of methods used to confirm the interaction would be beneficilal for the estimation of its relevance for proteins whose interaction was confirmed by a single group and recently identified interactors, so we added the information in the manuscript.
PGAM5 interacts with KEAP1 via the conserved NXESGE motif. This interaction has been validated by two different groups in several ways including affinity purification coupled with mass spectrometry and co-immunoprecipitation of overexpressed KEAP1 and endogenous PGAM5 (ref. 102, 103 in revised manuscript).
We have added the information about the confirmation of CDK20-KEAP1 interaction by co-immunoprecipitation of endogenous proteins from the paper Wang et al., 2017 (ref. 147) in the text.
We have also added the information about the methods used for the confirmation of the interaction with KEAP1 for all proteins described in the section 13. Other proteins.
FAM117B interaction was first identified by Hast et al. 2013 (ref. 90). Endogenous protein was co-immunoprecipitated with overexpressed KEAP1 and Kelch domain of KEAP1. Interaction was confirmed by Zhou et al. 2023 (ref. 156). They co-immunoprecipitated endogenous KEAP1 and FAM117B from HGC-27 and AGS gastric cancer cell lines, confirmed direct interaction with GST-pulldown of purified proteins and also measured the affinity of binding by microscale thermophoresis (MST). They determined the Kd of binding of NRF2 and NRF2ΔDLG to GST-KEAP1, as well as FAM117B. From what they wrote, it would seem that they measured the MST with full length proteins, however, there are no details about protein purification or purchase in the manuscript. The Kd that they determined for KEAP1-NRF2 interaction is around 5 µM, several orders of magnitude higher than previously published Kds obtained by ITC for the binding of NRF2 peptides to Kelch domain (around 20 nM), so it is very questionable if their binding affinity results are relevant.
OTUD1 interaction with KEAP1 was confirmed by co-immunoprecipitation of endogenous proteins by two different groups (ref. 131 and 132). OTUD1 was also identified in high-throughput screen of KEAP1 interactors conducted by Hast et al. 2013 (ref. 90).
DPP9-KEAP1 interaction was confirmed by co-immunoprecipitation of endogenous protein in several cell lines from clear cell renal cell carcinoma (786-O, CCF-RC1, Caki-1, and OSRC2) and kidney renal papillary cell carcinoma (Caki-2 and ACHN) (ref. 159). Interaction of endogenous DPP9 and KEAP1 was also confirmed in liver cancer cell lines (SK-Hep-1 and Hep G2) (ref. 160).
KEAP1-Gankyrin interaction was confirmed by co-immunoprecipitation of endogenous proteins in HEK293T cell line (ref. 161).
KEAP1-RMP interaction was confirmed by co-immunoprecipitation of endogenous proteins from cholangiocellular carcinoma cell line HuCCT1 (ref. 162).
- What are the affinity values of these proteins for KEAP1?
As far as we know, affinity values for the binding of competitive interactors to KEAP1 have been analyzed only for some proteins and primarily using peptides representing the ETGE loop of the competitive interactor proteins. Cino et al. (2013) employed isothermal titration calorimetry (ITC) and computational modeling to investigate the binding of 20-amino-acid-long peptides, corresponding to the ETGE motif loop of NRF2 and six competitive interactors (PALB2, PGAM5, WTX, FAC1, SQSTM1, and PTMA), to the Kelch domain of KEAP1. The study revealed large variations in peptide affinity for the Kelch domain, ranging from 23 nM for NRF2 to approximately 12 µM for PTMA-isoform 2 (an isoform missing Glu40, located two residues upstream of the ETGE motif in comparison to wild-type PTMA). PALB2, PTMA, and WTX exhibited a Kd in the 100–200 nM range, indicating they had approximately 5–10 times higher affinity for the Kelch domain compared to the lower-affinity DLGex site of NRF2 (approximately 1 µM). In contrast, PTMA, FAC1, and SQSTM1 demonstrated lower affinities for the Kelch domain (approximately 1 µM) [1].
We did not include the interaction with the nucleosome-remodeling factor subunit BPTF (Fetal Alz-50 clone 1 protein, FAC1) in this review, as the protein does not have ETGE-like motif and the interaction was not confirmed on endogenous proteins .
Our group utilized ITC to measure the affinity of 11- and 26-amino-acid DPP3 peptides to the Kelch domain and found Kd values of approximately 56 nM and 33 nM, respectively, positioning the DPP3 ETGE loop as one of the strongest competitive interactors of KEAP1 [2].
The data obtained from affinity measurements of short peptides cannot be directly extrapolated to cellular contexts. Apart from that, different groups employed different peptides and it is difficult to make a meaningful comparison of those results. Therefore, we decided not to add the data about affinity measurements to the paper. However, it is worth mentioning that peptides with higher affinity for NRF2 may serve as promising starting points for the development of modulators of NRF2 activity.
- Table 1 is not referenced in the main text and should be cited to ensure clarity and relevance.
Reference to the Table 1 was added to the text.
[1] E.A. Cino, R.C. Killoran, M. Karttunen, W.-Y. Choy, Binding of disordered proteins to a protein hub, Sci Rep 3 (2013). https://doi.org/10.1038/srep02305.
[2] S. Matić, I. Kekez, M. Tomin, F. Bogár, F. Šupljika, S. Kazazić, M. Hanić, S. Jha, H. Brkić, B. Bourgeois, T. Madl, K. Gruber, P. Macheroux, D. Matković-Čalogović, M. Matovina, S. Tomić, Binding of dipeptidyl peptidase III to the oxidative stress cell sensor Kelch-like ECH-associated protein 1 is a two-step process, Journal of Biomolecular Structure and Dynamics 39 (2021) 6870–6881. https://doi.org/10.1080/07391102.2020.1804455.
Reviewer 3 Report
Comments and Suggestions for Authors
The manuscript provides a comprehensive review of the KEAP1-NRF2 pathway, emphasizing its dual role in cancer progression and response to oxidative stress. The authors discuss competitive interactors and their implications for carcinogenesis, including potential therapeutic avenues. While the manuscript is informative, several areas require elaboration, improved organization, and additional references to enhance clarity and scientific rigor.
Detailed Comments
Abstract
Lines 10–15: Clarify the protective and oncogenic roles of NRF2 in different contexts, as the dual nature may be unclear to non-specialist readers.
Lines 20–25: Include specific examples of KEAP1 competitive interactors mentioned in the manuscript to strengthen the abstract's relevance.
Introduction
3. Lines 40–50: Provide updated statistics on cancer in general as well as this cancer type prevalence, including survival rates, to highlight the critical need for biomarkers and treatment target. Cite “Cancer statistics, 2024, 2024”. Then give intro in cancer therapy in general, cite NIH paper “Cancer treatments: Past, present, and future, 2024” (PMID: 38909530)for more information. Then emphasize cancer types with the highest prevalence of KEAP1 or NRF2 mutations.
4. Lines 60–65: Expand on the therapeutic challenges associated with targeting the KEAP1-NRF2 pathway and potential strategies for overcoming resistance. Mention recent method for drug resistance such as bioinformatic combined with CRISPR screening“CRISPR screening and cell line IC50 data reveal novel key genes for trametinib resistance, 2025” discussion how these help study the KEAP1-NRF2 pathway for overcoming resistance.
Materials and Methods
5. Lines 100–110: Explain how literature was selected for review, including criteria for inclusion/exclusion.
6. Lines 130–135: Justify focusing on the specific competitive interactors discussed and whether any notable interactors were excluded due to insufficient evidence.
Results and Discussion
Sequestosome-1 (SQSTM1/p62)
7. Lines 83–105: Discuss why SQSTM1 is particularly prominent in certain cancers (e.g., ovarian, lung) and whether its expression correlates with prognosis.
8. Line 120: Explain the therapeutic potential of targeting SQSTM1 in combination with NRF2 inhibitors.
Minichromosome Maintenance 3 (MCM3)
9. Lines 135–145: Elaborate on the coordination of DNA replication and redox homeostasis via MCM3-KEAP1 interactions.
10. Line 157: Include examples of cancers where MCM3 is most frequently overexpressed and associated with poor outcomes.
Partner and Localizer of BRCA2 (PALB2)
11. Line 180: Explain the implications of PALB2 interactions with KEAP1 for chemoresistance and oxidative stress adaptation in cancer cells.
Other Competitive Interactors
12. Lines 250–270: Highlight novel findings about interactors such as DPP9 and their role in ferroptosis resistance, and discuss their potential as therapeutic targets.
13. Lines 290–310: Discuss limitations in the current understanding of certain competitive interactors, particularly those lacking functional validation.
Figures and Tables
14. Table 1: Ensure the cancer types listed for each interactor are comprehensive and supported by citations.
15. Figures 2–4: Simplify complex schematics to improve readability for a broader audience.
16. Lines 610–630: Emphasize translational research opportunities, such as using KEAP1-interacting proteins as biomarkers for therapy selection or treatment monitoring.Dand discuss the potential bias.
Author Response
Thank you for the detailed and comprehensive review of our paper. Suggested revisions are going to increase the quality and relevance of our work. Please find the response in the attached file.

Round 2
Reviewer 2 Report
Comments and Suggestions for Authors
I recommend publishing the manuscript in its current version.
Reviewer 3 Report
Comments and Suggestions for Authors
ok